# Unsupervised Image-to-Video Adaptation via Category-aware Flow Memory Bank and Realistic Video Generation

## ABSTRACT

Image-to-Video adaptation is proposed to train a model using labeled images and unlabeled videos to facilitate the classification of unlabeled videos. The latest work synthesizes videos using still images to mitigate the modality gap between images and videos. However, the synthesized videos are not realistic due to the camera movements are only simulated in 2D space. Therefore, we generate realistic videos by simulating arbitrary camera movements in 3D scenes, and then the model can be trained using the generated source videos. Unfortunately, the optical flows from the generated videos have unexpected negative impacts, resulting in suboptimal performance. To address this issue, we propose the Category-aware Flow Memory Bank, which replaces optical flows in source videos with real target flows, and the new composed videos are beneficial for training. In addition, we leverage the video pace prediction task to enhance the speed awareness of the model in order to solve the problem that the model performs poorly in handling some categories with similar appearances but significant speed differences. Our method achieves state-of-the-art performance and comparable performance on three Image-to-Video benchmarks.

## CCS CONCEPTS

• **Computing methodologies → Activity recognition and understanding**.

## KEYWORDS

Image-to-Video Adaptation, Category-aware Flow Memmory Bank, Realitic Video Generation, Speed Awareness Enhancement, Action Recognition

**ACM Reference Format:**
Anonymous Author(s). 2024. Unsupervised Image-to-Video Adaptation via Category-aware Flow Memory Bank and Realistic Video Generation. In *Proceedings of the 32nd ACM International Conference on Multimedia (MM '24), October 28-November 1, 2024, Melbourne, AustraliaProceedings of the 32nd ACM International Conference on Multimedia (MM'24), October 28-November 1, 2024, Melbourne, Australia.* ACM, New York, NY, USA, 12 pages. https://doi.org/XXXXXXX.XXXXXXX

## 1 INTRODUCTION

Video recognition is currently an active research direction in the field of multimedia due to its wide-ranging applications, such as

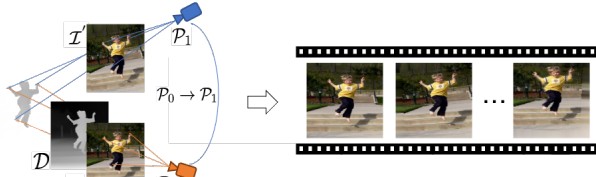

(a) Move camera along a virtual motion and generate a new view.  (b) Generated video.

**Figure 1: Generating a video from a single source image is achieved through the Depthstillation [1] pipeline. Initially, we project the pixels in the input image $I$ into 3D space, guided by the corresponding estimated depth map $\mathcal{D}$. Subsequently, we move the camera along a virtual motion path from $\mathcal{P}_0$ to $\mathcal{P}_1$. Finally, this process yields a new view $I'$. By combining these synthesized views, we can construct a more realistic source video.**

video retrieval [11, 42], intelligent video surveillance [33, 46], and video captioning [28, 32]. However, training a high-performance video classifier requires collecting and annotating a large amount of video data, which is costly and time-consuming. As images are easier to annotate than videos, and there are numerous labeled image datasets accessible, image-to-video domain adaptation methods [4, 21, 22, 30, 51] that leverage the labeled images and unlabeled videos for training high performance video classifier appear as a challenging task and attract much attention.

The first challenge of image-to-video adaptation is the modality gap between images and videos. This gap refers to the fact that the temporal information in videos does not exist in source images. Bridging this modality gap is necessary for transferring knowledge from source domain to target domain effectively. Another challenge arises from domain discrepancy caused by variations in scenes, image styles and so on between source images and target video frames. Domain discrepancy is a key factor causing models trained in the source domain to perform poorly in the target domain. Overcoming these two key challenges of modality gap and domain discrepancy is crucial for achieving effective image-to-video adaptation.

Existing approaches [47, 48] predominantly employ a two-stage paradigm to address the challenges of domain discrepancy and modality gap. The first stage involves frame-level adaptation to reduce domain discrepancies between source images and target video frames. The second stage entails learning a spatio-temporal model to bridge the modality gap and incorporate temporal information. For example, Wei et al. [21] first employs DANN [9] for frame-level alignment and then leverages pseudo-labels from the first stage for self-supervised learning on the target videos. Recently, Zhuo et al. [51] proposes a single stage method ST-I2V by synthesizing source videos from static images with the help of

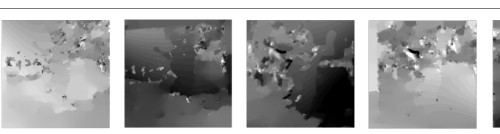

(a) Optical flows from generated source video.

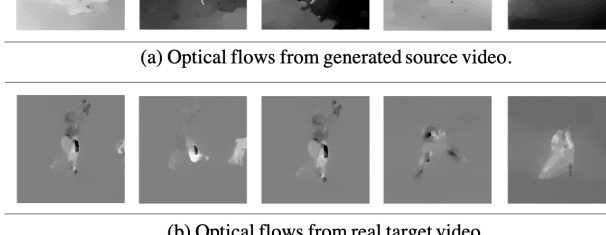

(b) Optical flows from real target video.

**Figure 2: We employ TV-L1 [49] to extract optical flows from both source and target videos for category 'jump', as shown in (a) and (b) respectively. It is evident that there is a significant discrepancy between the flows of the source and target videos. Specifically, the flows of the source video exhibit more interference and noise, while those of the target video appear cleaner.**

Grad-CAM [35], and solve the image-to-video adaptation problems with video-to-video domain adaptation methods.

Though being simple and effective, ST-I2V [51] randomly selects regions within image as intermediate frames to simulate camera displacement in 2D space which results in unrealistic synthesized video with improper temporal information. To address this issue, we rethink the imaging process in 3D space. As shown in Fig. 1, by recovering the position of camera, we can simulate the arbitrary movement of camera and generate realistic frame. Compared with ST-I2V [51], our method is simpler as it does not require training an additional classifier to locate major objects for an action. Besides, the complete original images are retained, avoiding any loss of crucial appearance information. The generated source video preserves static appearance and dynamic motion information, which is beneficial for training a discriminative spatio-temporal model.

Utilizing the generated source videos, we can train a simple but effective spatio-temporal baseline model through cross entropy loss with labels of source domain. However, we observe an unfavorable phenomenon that the optical flows extracted from generated videos are not helpful for training a discriminative model and even lead to suboptimal performance. As shown in Fig. 2, the optical flows from generated source videos exhibit more noise and interference compared with those of the target videos. Consequently, there is a noticeable distribution gap between the source and target videos, resulting in suboptimal performance.

To address the significant discrepancies between the flows in source and target videos, we construct a Category-aware Flow Memory Bank (CFB). The memory bank stores real flow data for each category within the real videos from target domain, where the category is determined by the pseudo label of target video. For a source video with ground-truth label $l$, we randomly select a flow of corresponding category $l$ from the CFB. Then the selected flow is used to replace the original flow in the source video. As the new source video is more similar to the target video, the performance of the model is greatly improved.

Nevertheless, it is still difficult for the trained spatio-temporal model to distinguish categories with similar visual appearances but significant differences in speeds, such as 'walk' and 'run'. So we leverage the video pace prediction task [41] to enhance the model's perception of speed by altering video playback speeds. Specifically, we sample video clips at varying pace rates and treat the pace rates as labels. Subsequently, the spatio-temporal model is also trained with cross-entropy loss with video playback speed labels. Additionally, integrating video playback speed prediction task prevents the model overfitting on the source domain and enables the model to better generalize to target domain.

We validate our method on three widely used image-to-video adaptation benchmarks. The experimental results show that our method performs favorably against the current state-of-the-art approaches. We achieve the best-published results on the challenging E→H and B→U benchmarks, and competitive results on the S→U benchmark. Ablation studies are presented to verify the contribution of each key component in our approach.

In a nutshell, our contributions are as follows:

- To generate realistic source videos, we simulate camera movements in a 3D scene and capture new camera views that serve as source video frames, inspired by [1]. The generated source videos are very promising in training a discriminative spatio-temporal model.
- We propose a Category-aware Flow Memory Bank (CFB) to compensate the improper temporal information of the generated videos in source domain. By replacing the original flows of source videos with the retrieved flows from CFB for training, a remarkable improvement in the performance of model is achieved.
- We integrate video pace prediction task [41] to enhance the model's perception of speed, which enables the model to distinguish categories with similar visual appearances but differences in speeds.
- Extensive experimental results show that our method achieves the best performance on the challenging E→H and B→U benchmarks and attains comparable results on the S→U benchmark.

## 2 RELATED WORK

**Image-to-video adaptation** methods focus on transferring knowledge from the image domain to the video domain. Existing unsupervised image-to-video adaptation tasks assume that only the labels of images are accessible, while the labels of the target videos are inaccessible. Mitigating the modality gap and reducing the distribution discrepancy are the primary objectives of image-to-video adaptation approaches [16, 20, 47]. For example, generative adversarial network (GAN) [12] is used to learn the mapping between image features and video features in HiGAN [48] and SymGAN [47]. The spatio-temporal causal graph [4] pursues similar goals. In order to mitigate the inherent modality gap between images and videos during domain adaptation, these methods leverage the strong generative modeling capabilities of GANs to transfer knowledge across modalities. CycDA [21] employs a four-stage method for adaptation. Class-agnostic alignment is performed in the first stage to derive pseudo-labels for training an independent spatio-temporal model in the second stage. The next two stages conduct iterative spatial alignment and spatio-temporal learning, with bidirectional knowledge transfer between the two components. Zhuo et al. [51] propose a new framework ST-I2V which synthesizes videos from

source static images, thereby converting the image-to-video adaptation task into video-to-video adaptation task. ST-I2V only simulates the transformation of the camera position in the 2D space, resulting in suboptimal performance. In contrast, we generate more realistic video frames by adjusting camera viewpoints at different positions in the 3D space. Additionally, we adopt both RGB and flow branches to construct our spatio-temporal model.

**Video-to-video adaptation.** Different from image-to-video adaptation, video-to-video adaptation methods are proposed to adapt labeled source videos to unlabeled target videos [6, 18, 25, 29], with their primary focus on addressing the challenges of domain alignment. Discrepancy-based methods are introduced to explicitly minimize domain discrepancies. For example, PTC [10] minimizes the Maximum Mean Discrepancy (MMD) [13] loss across both RGB and optical flow modalities to reduce the domain shift, resulting in improved performance. DVM [43] employs MixUp [50] to mitigate the domain-wise gap. This is achieved by progressively fusing the target videos with the source videos at the pixel-level, allowing for better alignment and adaptation between the domains. In our method, we propose a category-aware flow memory bank to replace the flow data in the generated videos in source domain, thereby reducing the domain gap.

**Video self-supervised learning** offers a promising annotation-free approach for representation learning in video domain. However, learning video representations is challenging due to temporal dynamics, motion, and other environmental factors. One key motivation behind defining pretext tasks is the idea that if a model can perform well on a complex task that requires a high-level understanding of video content, then it will learn more generalizable representations. For example, Jing et al. [15] and Wang et al. [40] design a task that applies appearance augmentations to video clips, and then the model is asked to classify the specific augmentation method that is applied. Fernando et al. [8] and Xu et al. [44] both design their approaches in a way that involves shuffling the order of video frames and having the model predict whether the video segment has been frame-shuffled. In order to enhance the model's awareness of speed and avoid overfitting problem in source domain, we introduce another simple video self-supervised learning method called video pace prediction [41].

## 3 METHODOLOGY

The goal of our method is to train a model that can achieves effective classification performance on target videos where the ground-truth annotations come from the labeled source image domain only. We train and evaluate our model in closed-set setting which means only the data from common categories in source and target domains will be used. It is supposed that there are a labeled source image domain $I_s = \{(x_i^s, y_i^s)\}_{i=1}^{n_s}$ and an unlabeled target video domain $V_t = \{v_j^t\}_{j=1}^{n_t}$. Both domains contain the same $C$ classes.

Our overall framework is shown in Fig. 3. Since a large modality gap exists between images and videos, we first convert the image-to-video task into a video-to-video task. To simply construct a spatio-temporal model, we employ the I3D [3] network pretrained on the Kinetics dataset [17] as the backbone for both the RGB and flow branches. Following the instructions of I3D, we extract optical flows from generated source videos and real target videos. The

extracted flows are denoted by $f_s$ in source domain and $f_t$ in target domain. We address the issue of suboptimal performance caused by utilizing the original flows from the generated videos, with our proposed CFB. Furthermore, we enhance the model's ability to perceive speed by applying video pace prediction task [41].

### 3.1 Source video generation

Although Zhuo et al. [51] have provided an effective method for source video generation, the generated videos may not be sufficiently realistic. This is because the approach only simulates the movement of the camera's viewpoint in 2D space, neglecting the fact that the body actions within the video should exist in 3D space.

To address this issue, we first generate source videos from source images via a virtual camera motion engine module, inspired by [1]. For a given source image $\mathcal{I}$, we employ MiDaS [34] to estimate the depth map $\mathcal{D}$. The estimated $\mathcal{D}$ is then utilized to project pixels in $\mathcal{I}$ into 3D space based on the inverse intrinsic matrix $\mathcal{M}^{-1}$ (the intrinsic matrix is used to transform 3D world coordinates into 2D image coordinates captured by a camera). Assuming that $\mathcal{I}$ is captured by the camera at 3D location $\mathcal{P}_0$, we apply an arbitrary virtual motion to the camera, moving it to a new position $\mathcal{P}_1$. Specifically, we generate a rotation matrix $\mathbf{R}$ and a translation vector $\vec{t}$ by sampling a random triplet of Euler angles and a random 3D vector, respectively. The transformation matrix is then defined as $\mathcal{T}_{0\to1} = (\mathbf{R}|\vec{t})$ which is corresponding to the virtual motion path $\mathcal{P}_0 \to \mathcal{P}_1$. For each pixel with coordinate $p$ in $\mathcal{I}$, the coordinate $p'$ of its corresponding pixel in $\mathcal{I}'$ acquired from the new viewpoint $\mathcal{P}_1$ can be obtained by:

$$p' \sim \mathcal{M}\mathcal{T}_{0\to1}\mathcal{D}(p)\mathcal{M}^{-1}p. \quad (1)$$

where $\mathcal{D}(p)$ is the depth value with coordinate $p$. Finally, the new image $\mathcal{I}'$ is obtained through forward warping.

Following these steps, we alternately generate the subsequent video frame by utilizing the previously generated frame, thereby generating a source video $v_i^s$ with continuous action. Subsequently, we utilize the TV-L1 algorithm [49] to compute optical flow $f_i^s$ from the i-th source video $v_i^s$ and optical flow $f_i^t$ from the i-th target video $v_i^t$, respectively. Therefore, the source domain is denoted by $V_s = \{(v_i^s, f_i^s, y_i^s)\}_{i=1}^{n_s}$, and the target domain is denoted by $V_t = \{(v_i^t, f_i^t)\}_{i=1}^{n_t}$.

Given the source video $(v_i^s, f_i^s, y_i^s)$, we use the cross-entropy loss as the classification loss at the frame-level and video-level referring to I3D [3] and ST-I2V [51]. The frame-level classification entropy loss named local loss $\mathcal{L}_{loc}$ is defined as below,

$$\mathcal{L}_{loc} = \frac{1}{B} \sum_{i=1}^{B} \sum_{k=1}^{K} CE(\hat{p}(v_i^s)_k, y_i^s). \quad (2)$$

Here, $CE(\cdot, \cdot)$ represents the cross-entropy loss, and $\hat{p}(v_i^s) \in \mathcal{R}^{K \times C}$ corresponds to the network's logits over $K$ RGB frames from a generated video $V_i^s$. The variable $B$ denotes the batch size.

Following the instructions in I3D [3], we train the RGB branch and the flow branch individually. So the video-level classification entropy loss $\mathcal{L}_c$ is defined as below,

$$\mathcal{L}_c = \frac{1}{B} \sum_{i=1}^{B} (CE(\overline{\hat{p}(v_i^s)}, y_i^s) + CE(\overline{\hat{p}(f_i^s)}, y_i^s)), \quad (3)$$

**Figure 3: The overall framework of our approach begins by generating videos from static images for source domain. Subsequently, we replace the flow input of the generated source video with the retrieved flow data from the proposed category-aware memory bank. We sample a new video segment from the target video, for instance, by applying a 2× speedup (i.e. VP=2) from the original video clip. Then, the RGB frames and the optical flows are fed input into the RGB branch and flow branch separately. Finally, the representations from source domain are used to compute the cross-entropy losses for classification (i.e. CE Loss). And the representations from target domain are used to calculate the cross-entropy loss for video pace prediction task (i.e. VC Loss).**

where $\overline{\hat{p}(v_i^s)} \in \mathcal{R}^{K \times C}$ is an average over logits of $K$ RGB frames from video $V_i^s$ and $\overline{\hat{p}(f_i^s)} \in \mathcal{R}^{K \times C}$ is an average over logits of $K$ flows from video $V_i^s$.

During the inference phase, the class logits $\overline{\hat{p}(v_i^s)}$ and $\overline{\hat{p}(f_i^s)}$ obtained from both RGB and flow branches are normalized via softmax activation function. The normalized results are then summed together, yielding probabilities for each action category. Finally, the action category with the highest probability is selected as the predicted category for the current video sample.

### 3.2 Category-aware flow memory bank

As shown in Fig. 2, the optical flow frames extracted from generated source videos contain more interference in comparison to the cleaner optical flow frames present in target videos. This phenomenon indicates the significant distribution gap between the source and target domains.

To address the negative impacts of the original flows from source videos on the performance of model, we propose a Category-aware Flow memory Bank (CFB). As shown in Fig. 3, we train the model using the original source videos $v_s$ and $f_s$ during the warm-up epochs. After the warm-up phase, we utilize the trained model to assign pseudo label for each target video sample. Subsequently, we sort the samples based on the confidences of these pseudo-labels and retain only the top $N$ samples for each pseudo category. In this manner, we construct a memory bank of size $C \times N$, where $C$ represents the number of categories. We denote the $n$-th flow of category $c$ as $\mathcal{F}_c^n$.

Given the $i$-th generated video sample $V_i^s = (v_i^s, f_i^s, y_i^s)$ in source domain, we replace the original flow $f_i^s$ in $V_i^s$ with a randomly selected flow $\mathcal{F}_c^n$, resulting in a new video sample $\hat{V}_i^s = (v_i^s, \mathcal{F}_c^n, y_i^s)$ where $c$ is equal to $y_i^s$. Then the Eqn. (3) is modified as below,

$$\mathcal{L}_c' = \frac{1}{B} \sum_{i=1}^{B} (CE(\overline{\hat{p}(v_i^s)}, y_i^s) + CE(\overline{\hat{p}(\mathcal{F}_c^n)}, y_i^s)), \qquad (4)$$

where $\overline{\hat{p}(\mathcal{F}_c^n)} \in \mathcal{R}^{K \times C}$ is an average over logits of K flows of selected flow $\mathcal{F}_c^n$.

### 3.3 Speed awareness enhancement

Distinguishing categories with similar visual appearances but significant differences in speeds, such as 'walk' and 'run', poses a challenge for the model. To address this issue, we leverage video pace prediction task [41] to empower the model with the capability to perceive speed by altering video playback speeds. When provided with a video in its natural pace containing $K$ frames, we sample video segments $\widetilde{v}_t$ by various video pace rates $r$. These pace rates correspond to labels from a predefined pace label space $\mathcal{R}^{C_{vp}}$. For example, we produce three pace rate candidates {normal, fast, super fast}, where the corresponding pace labels $r$ are {1, 2, 3}, respectively. We randomly choose the starting frame over $K$ frames for each target video and then loop over the video at a regular interval $r$ until we obtain the desired number of frames for training.

With the video segment $\widetilde{v}_t$ which is sampled by pace rate $r$, the objective of pace prediction task is to understand the content of

the video segment and predict the correct pace rate. Subsequently, we utilize the video pace labels to train our model using the cross-entropy loss $\mathcal{L}_{vc}$, which is defined as follows:

$$\mathcal{L}_{vc} = \frac{1}{B} \sum_{i=1}^{B} CE(\hat{r}(\widetilde{v}_i^t), r_i), \tag{5}$$

where $r_i$ denotes the video pace label of $i$-th video, $\hat{r} \in \mathcal{R}^{C_{vp}}$ is the predicted pace logits and $C_{vp}$ denotes the size of pace label space.

Overall, all loss functions mentioned above form the complete objective:

$$\mathcal{L} = \mathcal{L}'_c + \lambda_1 \mathcal{L}_{loc} + \lambda_2 \mathcal{L}_{vc}, \tag{6}$$

where $\lambda_1$ and $\lambda_2$ are trade-off parameters.

## 4 EXPERIMENTS

### 4.1 Datasets and setup

We evaluate our method through experiments on three standard image-to-video adaptation benchmarks: E→H, B→U and S→U. In the case of the E→H, we utilize the EADs [5] dataset, which comprises Stanford40 [45] and the HII dataset [37], as the source image domain, and HMDB51 [19] as the target video domain. There are 13 common classes between EADs and HMDB51 for image-to-video adaptation. The labeled source images and the unlabeled target videos are used to train a model. Regarding B→U, we employ the BU101 [26] dataset as the labeled source image domain and UCF101 [36] as the unlabeled target video domain. We use a total of 101 classes for the image-to-video adaptation task, as the classes in BU101 completely correspond to those in UCF101. For the S→U benchmark, we substitute the source image domain from the B→U benchmark with the Stanford40 [45] dataset. To perform the image-to-video adaptation task, the 12 common classes between Stanford40 and UCF101 are selected for training and evaluation.

### 4.2 Implementation details

To generate videos for source domain, we utilize MiDaS [34] whose backbone is pretrained BEIT-Large-512 [2] to extract depth maps from still images in source domain. Subsequently, we utilize Depth-stillation [1] to generate 16 video frames using the extracted depth maps and still images. The coefficient of translation vector $\vec{t}$ is set to 0.01. Some generated frames can be found in Supplementary material. We use all 16 frames of the generated source videos during training following ST-I2V [51] for fair comparison.

For constructing a category-aware memory bank with high-quality pseudo-labels, we train the model for 10 epochs as warm-up phase. We then employ the model to assign pseudo-labels for target videos in each subsequent training epoch. We select the top $N = 60$ samples with the highest confidence for each category based on the confidences of the pseudo-labels and store their flow data in the memory bank. The influence of number of selected samples $N$ can be found in Parameter sensitivity analysis of subsection 4.4.

For building a spatio-temporal model, we use I3D model [3] with both RGB and flow branches pretrained on the Kinetics dataset [17]. We replace the last classifier layer with a fully connected layer that includes $C$ neurons. We freeze the first three Unit3D blocks following ST-I2V [51] to accelerate the training process.

We train the model with mini-batch stochastic gradient descent optimizer where the momentum and weight decay are set to 0.9 and 0.0001, respectively. The initial learning rates, batch sizes and total epochs are set to (0.05, 0.1, 0.015), (16, 32, 32) and (60, 30, 20) for E→H, B→U and S→U benchmarks, respectively. We also adopt multistep decaying learning rate with a 0.1 decay rate where the milestones are half of the total epochs and the 2/3 of the total epochs. After the warm-up training phase, CFB is applied when the pseudo labels of target videos are more accurate. Following ST-I2V [51], the values of hyper-parameter $\lambda_1$ are set to (1, 20, 1) for E→H, B→U and S→U benchmarks, respectively.

We set the size of video pace label space to 5 for all benchmarks, which includes five video pace labels {1, 2, 3, 4, 5}. We randomly select a beginning frame for each target video and loop over the video at the generated video pace rate until the training video clip contains 16 frames. The generated video pace rate is treated as the pace label. For example, if there are 30 frames in the video, and loop over it starting from the 20th frame at 2× speed, the indices of the selected frames are {20, 22, 24, 26, 28, 30, 2, 4, 6, 8, 10, 12, 14, 16, 18, 20}. And the 2× pace is regarded as pace label. During inference, we follow the approach of ST-I2V [51] and extract 32 frames uniformly from each target video for fair comparison. The values of hyper-parameter $\lambda_2$ are set to (0.2, 0.001, 0.01) for E→H, B→U and S→U benchmarks, respectively.

### 4.3 Competitors and results

In our experiments, we conduct comparative evaluations against several prevailing approaches: The DANN [9] pioneers domain adversarial training for classical image-level adaptation. JAN [24] reduces the image-level domain shift by aligning the joint distributions of multiple domain-specific layers. DAL [27] is another image-level domain adaptation method that introduces a novel domain adaptation layer to align source and target distributions with a reference distribution. MEDA [39] minimizes structural risk to train a domain-agnostic classifier on the Grassmann manifold and dynamically aligns the distributions of multiple domains while evaluating the significance of marginal and conditional distributions. HiGAN [48] and SymGAN [47] attempt at bridging the modal gap by mapping image embeddings to video space using GAN [12]. DANN+I3D baseline leverages DANN adapted image features to train an I3D architecture with pseudo-labels which is implemented by Lin et al. [21]. CycDA [21] is a four-stage method that reduces domain discrepancies by using both class-agnostic and class-aware domain alignment techniques. It also utilizes pseudo labels to train a I3D model, effectively bridging the modality gap. ST-I2V [51] is the recent state-of-the-art approach, which employs Grad-CAM [35] and an additional classifier to generate source videos. It transforms the image-to-video domain adaptation task into a video-to-video domain adaptation task. Additionally, for reference, we include the lower bound (SO (Img), where SO stands for Source-only.) and the upper bound (ground truth supervised target) from works [21, 51].

In Tab. 1, we present comparison results. Our approach achieves new state-of-the-art performances on the E→H and B→U benchmarks and demonstrates comparable result on the S→U benchmark. Specifically, our method outperforms ST-I2V [51] by 6.1% and 2.2% on the E→H and B→U benchmarks, respectively. It's important to

Table 1: Results on E→H, B→U and S→U, averaged over 3 random trials.

| method | E→H | B→U | S→U |
|---|---|---|---|
| SO (Img) | 37.2 | 54.8 | 76.8 |
| DANN [9] | 39.6 | 55.3 | 80.3 |
| JAN [24] | 40.9 | - | 91.4 |
| HiGAN [48] | 44.6 | - | 95.4 |
| DAL [27] | 45.5 | - | 97.6 |
| MEDA [39] | 43.1 | - | 94.3 |
| SymGAN [47] | 55.0 | - | 97.7 |
| DANN+I3D | 53.8 | 68.3 | 97.9 |
| CycDA [21] | 62.0 | 72.6 | **99.1** |
| ST-I2V [51] | 71.3 | 78.9 | 98.6 |
| Ours | **77.4** | **81.1** | 97.3 |
| supervised target | 83.2 | 93.1 | 99.3 |

Table 2: Ablation study results on E→H, B→U, and S→U, averaged over 3 random trials.

| method | E→H | B→U | S→U |
|---|---|---|---|
| SO ([51]) | 59.0 | 60.2 | 96.3 |
| SO (RGB) | 59.8 | 76.4 | 96.6 |
| SO (RGB+flow) | 60.7 | 74.6 | 96.3 |
| SO (RGB+flow) + CFB | 74.1 | 80.2 | 97.2 |
| Full Model | **77.4** | **81.1** | **97.3** |

note that the E→H benchmark is considerably more challenging than B→U and S→U, given the difficulties in distinguishing categories within the HMDB51 dataset. Nevertheless, the performance of our model on the S→U benchmark still lags behind the current state-of-the-art method. This gap may arise from the fact that our approach primarily focuses on enhancing temporal information, while S→U benchmark relies less on temporal information which can be verified from the superior performance of SO (RGB) in Tab. 2. On the other hand, we can adopt some existing domain adaptation techniques like BNM [7] and MCC [14] to further improve our model on S→U benchmark, achieving new state-of-the-art result, as shown in Tab. 4.

The experimental results indicate the effectiveness of our source video generation method, the proposed category-aware flow memory bank and the speed awareness enhancement approach, in the context of image-to-video domain adaptation learning.

### 4.4 Ablation study

To study the contribution of each component in our approach towards the overall performance, we conduct the ablation study of our proposed approach. We evaluate the following variants of model: (1) **SO (RGB)**, which denotes the model that contains RGB branch only and is trained with the labeled generated source videos. (2) **SO (RGB+flow)**, the model is trained simultaneously using both the RGB branch and the flow branch with source data. (3) **SO (RGB+flow) + CFB**, is the model trained by replacing the original

flow data in source generated video samples with the retrieved flow data from CFB after the warm-up phase. (4) **Full Model**, that is trained with incorporating VPT (video pace prediction task) into the baseline model SO (RGB+flow) + CFB. Additionally, we include the performance of SO ([51]) for reference, which is trained with synthesized videos generated by Zhuo et al. [51].

The ablation study results are shown in Tab. 2. Comparing with SO ([51]), we can observe that our model SO (RGB), when using only the RGB branch (the same as SO ([51])), has brought improvements of 0.8%, 16.2% and 0.3% on E→H, B→U and S→U, respectively. This means that our video generation approach is more effective to learn a discriminative spatio-temporal model against ST-I2V [51]. When training the flow branch using the original flow data from source videos, it still provides some improvements on the E→H benchmark but has negative effects on S→U and B→U benchmarks. This phenomenon demonstrates the improper flow from source video is one of the key factors contributing to the poor performance of spatio-temporal models.

Our proposed Category-aware Flow Memory Bank (CFB) aims to address this issue. The results of SO (RGB+flow) + CFB show that our approach brings performance improvements across three benchmarks, is specially with gains of 13.4% and 5.6% in E→H and B→U benchmarks, respectively. After enhancing the model's perception of speed by introducing VPT, the performance of Full Model is further improved, validating the effectiveness of our approach. **Parameter sensitivity analysis.** To evaluate the parameter sensitivity, we conduct a series of experiments on the E→H benchmark. Fig. 4 reports the results of parameter sensitivity analysis, and more results can be found in Supplementary material.

The weight $\lambda_1$ is one of the key factors that influences the performance of the model. We evaluate the impact of different values of $\lambda_1$ using the baseline SO (RGB) and present the results in Fig. 4 (a). As observed, a small $\lambda_1$ leads to the model ignoring appearance characteristics of video frames, while a large $\lambda_1$ results in the model excessively focusing on appearance characteristics at the expense of temporal features. This further hinders the model's ability to comprehend the content of the video.

An appropriate dimension of CFB is a critical factor in determining the performance of model. So we investigate the impact on performance by setting different numbers of flow samples $N$ for each category in baseline SO (RGB+flow) + CFB and the results are shown in Fig. 4 (b). It is evident that the value of $N$ can significantly influences the performance of model. An excessively large $N$ may introduces noisy flow samples with low-quality pseudo labels, which leads to inferior performance. Conversely, a small $N$ may leads to insufficient model generalization.

To investigate the proper settings for VPT, we explore different video pace prediction loss weights $\lambda_2$ and sizes of video pace label space $C_{vp}$ based on baseline SO (RGB+flow) and the results are shown in Fig. 4 (c) and Fig. 4 (d) respectively. From the above experimental results, it is observed that employing video pace prediction task on SO (RGB+flow) can achieves competitive results under a appropriate range of $\lambda_2$ values. Additionally, the small size of the video pace label space restricts the model's ability to perceive speed. On the other hand, a over-sized video pace label space increases the difficulty of the speed prediction task, making it challenging for the model to learn meaningful semantic representations.

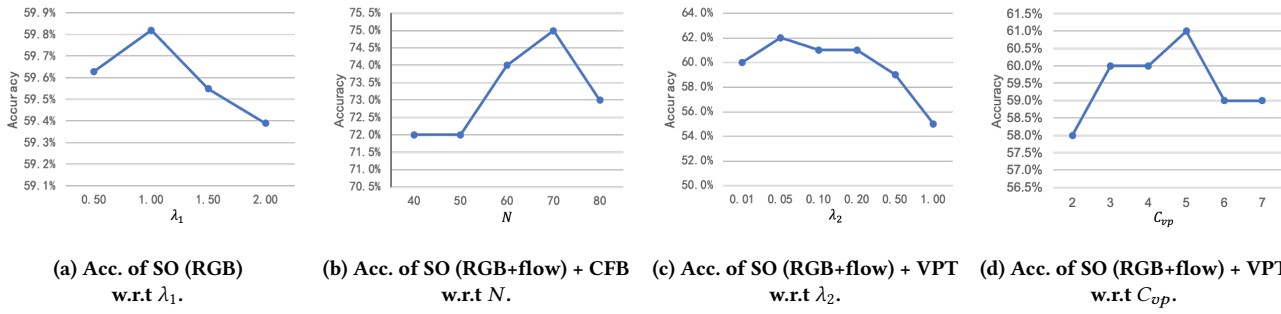

(a) Acc. of SO (RGB)
w.r.t $\lambda_1$.

(b) Acc. of SO (RGB+flow) + CFB
w.r.t $N$.

(c) Acc. of SO (RGB+flow) + VPT
w.r.t $\lambda_2$.

(d) Acc. of SO (RGB+flow) + VPT
w.r.t $C_{vp}$.

Figure 4: The plots of parameter sensitivity analysis. We obtain the results on E→H benchmark.

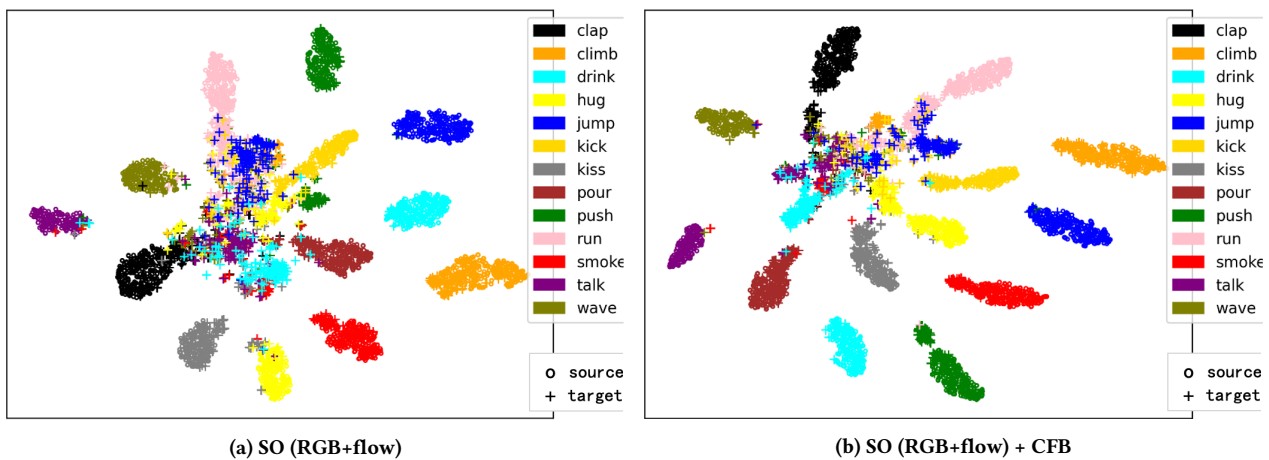

(a) SO (RGB+flow)

(b) SO (RGB+flow) + CFB

Figure 5: t-SNE visualizations of video representations (colored w.r.t. ground truth) from source and target domain in E→H benchmarks. We plot the representations of SO (RGB+flow) (a) and the representations of SO (RGB+flow) + CFB (b). We use 13 different colors to represent each category. We use 'o' and '+' to represent source representations and target representations respectively.

Table 3: Accuracies of different aggregation methods on E→H, averaged over 3 random trials.

| method | E→H |
| --- | --- |
| SO (RGB+flow) | 60.7 |
| Mean 60 | 59.0 |
| SimW Sum60 | 59.6 |
| SimW Mean top5 | 63.8 |
| SimW Sum top5 | 64.0 |
| SimW top1 | 72.5 |
| SimW Random 1/top5 | 73.2 |
| Random 1/60 | **74.1** |

**Different aggregation methods for retrieved flows.** Based on SO (RGB+flow) + CFB, we conduct a study to investigate the impact of different retrieved flow aggregation methods on the performance of model. We conduct these experiments on E→H benchmark. The results are presented in Tab. 3. After constructing a CFB which stores top 60 flows with the highest confidence for each category,

we first investigate the impact of aggregating all retrieved flows using mean pooling (denoted by Mean 60) on the performance of model. Next, we design various aggregation methods that automate the selection of retrieved flows through the cosine similarity of source and target RGB features. The cosine similarity is considered as the weight of each retrieved flow data. And we represent these methods using 'SimW' as the prefix. We evaluate the following methods: (1) **SimW Sum60**, means that we perform a weighted summation of all retrieved flows by using the weights assigned to each flow. (2) **SimW Mean top5**, we perform mean pooling on the top 5 retrieved flows with the highest weights. And then replace the source flow with the pooled flow. (3) **SimW Sum top5**, which denotes that we perform weighted summation of top 5 retrieved flows with the highest weights. (4) **SimW top1**, which means that we only choose the retrieved flow which own the highest weight. (5) **SimW Random 1/top5**, we randomly select one flow from the top 5 retrieved flows with the highest weights. At last, Random 1/60 represents the same aggregation setting as SO (RGB+flow) + CFB in Tab. 2, which is the one we used in the manuscript. For reference, we also include the result of SO (RGB+flow).

**Table 4: Accuracies on E→H and S→U after combining with DA Method, averaged over 3 random trials.**

| method | E→H | S→U |
|---|---|---|
| SO (RGB+flow) + CFB | 74.1 | 97.2 |
| SO (RGB+flow) + CFB + DAN | 74.7 | 97.3 |
| SO (RGB+flow) + CFB + MCC | **75.0** | 99.2 |
| SO (RGB+flow) + CFB + BNM | 74.5 | **99.3** |

The results indicate that regardless of using mean pooling or weighted summation to aggregate retrieved flows, the constructed flows lead to suboptimal results or even have significant negative impacts on the performance of model. This could be due to that the aggregated flows lose too much information and dissimilar to real flow data. With less (top5) flows for aggregation, the performance is improved. With only top1 flow, the performance is further improved. So we only use 1 flow without aggregation. We use Random 1/60 as we think that randomly choosing 1 flow may boosts the robustness of the model and the experimental result verifies our conjecture. So we use Random 1/60 in our manuscript.

## 4.5 Further remarks

**Integrating domain adaptation techniques.** We employ several typical domain adaptation techniques into the constructed baseline SO (RGB+flow) + CFB, including DAN [23], MCC [14] and BNM [7]. Specifically, we use $\mathcal{L}' = \mathcal{L}'_c + \lambda_1 \mathcal{L}_{loc} + \lambda_3 \mathcal{L}_{tf}$ to train the model. $\mathcal{L}_{tf}$ is the transfer loss likes MMD [23] loss, and BNM [7] loss.

The values of hyper-parameter $\lambda_3$ are set to (0.05, 0.3) for E→H and S→U, respectively. The results on E→H and S→U are shown in Tab. 4 and we also report the results of SO (RGB+flow) + CFB for better demonstration.

It is observed that the performance of our model can still be greatly improved by applying typical domain adaptation methods on our constructed video-to-video domain adaptation baseline which involves generating source videos and utilizing CFB. With recent state-of-the-art domain adaptation techniques MCC [14] and BNM [7], our method outperforms CycDA [21] on S→U benchmark, achieving new state-of-the-art performance. In addition, the results also demonstrate that the combination of our method with DA techniques shows good adaptability on E→H.

**Compatibility with CLIP.** We replace the network of RGB branch in sec. 4.2 with visual encoder of CLIP [31] whose backbone architecture is VIT-B/32. Different from the experimental hyper parameter settings described in sec. 4.2, we set total epochs to (40, 20) and the number of warm-up training phase to (10, 5) for E→H and S→U benchmarks respectively. And setting the batch size to 8, the learning rate of CLIP to 5e-5 and the prompt for text encoder to 'a video of a person {}.' for both benchmarks. {} in prompt indicates category names like 'climb', 'run' and so on.

For reference, we include the results of our approach employing the I3D network in RGB backbone. The results shown in Tab. 5 indicates that the performance improvements can be attributed to the robust representation learning capability and the powerful knowledge base of CLIP itself. By integrating CLIP into our method,

**Table 5: Accuracies on E→H and S→U after integrating with CLIP, averaged over 3 random trials.**

| RGB backbone | method | E→H | S→U |
|---|---|---|---|
| I3D | SO (RGB) | 59.8 | 96.6 |
| | Full Model | 77.4 | 97.3 |
| CLIP | SO (RGB) | 66.8 | 97.7 |
| | Full Model | **78.0** | **98.3** |

we can leverage its knowledge and capabilities to enhance the performance. Additionally, the compatibility between our method and CLIP is crucial for achieving further performance gains. The ability of our method to effectively incorporate CLIP's features and merge them with the existing framework allows for a synergistic effect, especially resulting in improved performance on E→H benchmark. Our approach's compatibility with prevailing large multimodal models like CLIP showcases its strength and demonstrates its ability to achieve better results.

**T-SNE visualization.** We visualize the representations of the baseline models, SO (RGB+flow) and SO (RGB+flow) + CFB, in Fig. 5 using t-SNE [38]. We project the source and target videos of E→H benchmark into 2-dimensional representations. Intuitively, there are more confusions among the representations from SO (RGB+flow) (Fig. 5 (a)), which can be improved by incorporating CFB, as shown in Fig. 5 (b). Specifically, for categories like 'pour', 'kick', and 'push', the incorporation of CFB enables the model to learn more discriminative representations, enabling better differentiation from others.

Moreover, for categories of 'talk', 'smoke', 'climb' and 'wave', the representations from both the source and target domains become closer after replacing the flows retrieved from CFB, as depicted in Fig. 5 (b). It is suggested that our proposed CFB further reduces the distribution discrepancies between source and target domains, which is beneficial for training a model with great generalization.

## 5 CONCLUSION

We overcome the challenges of image-to-video domain adaptation task, aiming to enhance the spatio-temporal model's discriminative ability for unlabeled video classification in the target domain. To mitigate the modality gap between labeled source images and unlabeled target videos, we generate realistic source videos by simulating diverse camera movements in 3D scenes and the new perspectives are served as frames. To further mitigate the negative influences of the flows extracted from the generated source videos, we propose the category-aware flow memory bank (CFB). By replacing the optical flow in a generated source video with real target flow which is retrieved from CFB, we create a new video sample that closely resembles the target video. Additionally, we leverage the video pace prediction task to enhance the model's perception of speed. Our proposed method demonstrates promising results compared with the current state-of-the-art approaches. In our current method, the domain discrepancy is not fully concerned, which could be further improved with video-to-video domain adaptation methods in the future work.

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
