# OpenReview forum: "Unsupervised Image-to-Video Adaptation via Category-aware Flow Memory Bank and Realistic Video Generation"
_acmmm.org/ACMMM/2024/Conference — MM2024 Poster_

### Official Review · Reviewer_rg6W · 2024-05-19

**Rating:** 4
**Confidence:** 2

**Summary:**

Image-to-Video adaptation refers to training models using labeled images and unlabeled videos to classify unlabeled videos. Existing research uses still images to synthesize videos, however, simulating camera motion in 2D space leads to synthesized videos that are not realistic. Therefore, this manuscript generates realistic videos by simulating arbitrary camera motions in 3D scenes, and then trains the model using the generated source videos.

**Strengths:**

1. This manuscript provides effective solutions to address the challenges of image-to-video adaptation, including the large modality gap between labeled source images and unlabeled target videos, the negative influences of extracted flow from generated source videos on model performance, and the difficulty in distinguishing categories with similar visual appearance but differences in speed. The experimental results show that the proposed method achieves state-of-the-art and comparable performance on three Image-to-Video benchmarks.
2. The manuscript has strong logical coherence and the effectiveness of the proposed method has been verified and analyzed through sufficient ablation experiments.

**Limitations:**

I am not very familiar with this field, but I believe that there are still the following issues that need to be clarified in this manuscript.
1. In Figure 4(b), the proposed model achieves optimal performance when N is set to 70. Why was it ultimately set to 60?
2. Why is the performance on benchmark B->U not reported in Tables 4 and 5?

**Suitability:**

2

---

### Official Review · Reviewer_ydNo · 2024-05-20

**Rating:** 4
**Confidence:** 1

**Summary:**

This paper proposes a new image to video adaptation method. First, it generates more realistic videos to mitigate the modality gap between source images and target videos. Then, to mitigate the gap between the flows extracted from the generated videos and target videos, it then proposes the category-aware flow memory bank, by replacing the optical flow in a generated source video with the optical flow selected from target videos, and then generate new composed videos for training. They also leverage the video pace prediction task to enhance the model's perception of speed. The proposed method achieves state-of-the-art performance in the field of image to video adaptation.

**Strengths:**

1. The proposed several approaches, such as the diverse camera movements simulation and the category-aware flow memory bank, are effective for unsupervised image to video adaption with good illustrations.
2. The proposed method achieves superior performance in the field of unsupervised image to video adaption.
3. The overall structure of this paper is well organized.

**Limitations:**

1. Is it possible for the authors to report the performance of "SO [51] + flow" and "SO (RGB) + CFB" to better demonstrate the effectiveness of the proposed category-aware flow memory bank?
2. It seems that the generated videos with 3D scenes are only used during the warm-up epochs if the category-aware flow memory bank will be used. If we could approach other pretrained image to video adaption models, are there any ways to directly use a category-aware flow memory bank to generate videos for training?

**Suitability:**

3

---

### Official Review · Reviewer_tEyZ · 2024-05-22

**Rating:** 4
**Confidence:** 2

**Summary:**

This manuscript aims to generate realistic videos by simulating camera movements in 3D scenes. To solve the interference and noise problem exhibited in the flows of the source videos, the authors introduce the Category-aware Flow Memory Bank, which replaces optical flows in source videos with real target flows. To solve the problem of identifying different categories of videos with similar appearance, the authors leverage the video pace prediction task to enhance the speed awareness.

**Strengths:**

1. For the task of image-to-video adaptation, the authors propose to simulate arbitarary camera movements in 3D scenes, which can provides more spatial-temporal information for the realistic video generation.
2. The authors propose a Category-aware Flow Memory Bank (CFB) to compensate the improper temporal information of the generated videos in source domain, reducing the improper impacts caused by the optial flow.
3. The writing of this paper is clear, and readers can understand the key points and effectiveness of the proposed method easily.

**Limitations:**

1.	The idea of simulating the camera movement in 3D scenes to generate video lacks novelty. The manuscript leverages a method called dephtstilled mentioned in "Learning optical flow from still images. CVPR'21". This method learns to obtain new perspective images and optical flow by giving a single image and the corresponding depth map, which contributes more to the cleanliness of the optical flows of the generated videos.
2.	Figure 2 shows the optical flow comparison between the source domain and the target domain, but it does not seem to indicate the generation results of the source domain video from which method, and it is doubtful whether the "interference and noise" problem is common.
3.	It is doubtful whether the proposed CFB strategy is reasonable. The authors claim that the Memory Bank stores the optical flow information extracted from the real target video and serves as pseudo labels for the video in the corresponding category of the source domain. Due to the similarity of optical flow in the same category (Line 172), the authors replace the optical flow of the source domain video in training. I think the judgment that the same category is equivalent to optical flow similarity is not rigorous enough. On the one hand, there may be large differences in optical flow within the same category, and the optical flow content between any two videos is misaligned. There may also be large similarities in optical flow between categories. On the other hand, there is a lack of sufficient experimental analysis to support the author's hypothesis.
4.	To deal with difficult similar video action categories, the authors sample video clips at varying pace rates and treat the rates as the video playback speed (line 539). I think it lacks rationality. For example, the difference between the two actions of "walk" and "run" is the size of the body movements, not the video playback speed. Moreover, the method proposed by the author appeared in "Self-supervised Video Representation Learning by Pace Prediction, ECCV2020".
5.	The effectiveness of CFB shown in Figure 5 is not intuitive, and the representations of a considerable number of categories are still difficult to distinguish after using the CFB strategy.

**Suitability:**

3

---

### Official Review · Reviewer_hBnA · 2024-05-24

**Rating:** 3
**Confidence:** 2

**Summary:**

This paper aims to tackle the task of Image-to-Video Adaptation in an unsupervised manner, which is based on the previous work and thus is incremental.

**Strengths:**

The paper writing is well organized and the quantitative result looks promising.

**Limitations:**

Author utilizes optical flow values retrieved from the Category-aware Flow Memory Bank to replace optical flow in the generated source video. I question the reasonableness of this design since optical flow should be data-dependent. It is unclear to me how replacing the optical flow values from one video with those from another can be rational. Optical flow represents the actual movement of specific positions in a video, and substituting these values does not seem to make sense. For this design to be effective, the source and target videos would need to exhibit similar motions in both direction and magnitude, which is not commonly observed.

**Suitability:**

1

---

### Meta-Review · Area_Chair_Dx5u · 2024-07-01

**Recommendation:** Accept (Poster)
**Confidence:** 4

**Metareview:**

This paper introduces a novel approach for unsupervised image-to-video adaptation using category-aware flow memory banks and realistic video generation techniques. Despite concerns raised about the rationale behind optical flow replacement and the novelty of some methodologies, the authors provided satisfactory responses that addressed these issues. As a result, all reviewers reached a consensus to accept this paper.